# Emotion Regulation Ability: Test Performance and Observer Reports in Predicting Relationship, Achievement and Well-Being Outcomes in Adolescents

**DOI:** 10.3390/ijerph18063204

**Published:** 2021-03-19

**Authors:** Zorana Ivcevic, Catherine Eggers

**Affiliations:** 1Yale Center for Emotional Intelligence, Yale School of Medicine, Yale University, New Haven, CT 06511, USA; 2Department of Psychology, Kenneth P. Dietrich School of Arts and Sciences, University of Pittsburgh, Pittsburgh, PA 15260, USA; cae70@pitt.edu

**Keywords:** emotion regulation ability, observer reports, emotional intelligence, academic achievement, relationship quality, affective well-being

## Abstract

This paper examines emotion regulation defined as one of the components of emotional intelligence ability and tests how emotion regulation predicts academic achievement, relationship quality, and affective well-being outcomes in adolescents. Specifically, we examine two ways of measuring emotion regulation ability—using performance test scores and through knowledgeable informant observations (teachers). While previous research supports the predictive validity of performance on ability tests of emotion regulation observer reports of emotion regulation have not received much empirical attention. In a sample of high school students, we test whether performance-tested and observer-assessed emotion regulation ability predict a range of outcomes beyond the Big Five personality traits and gender and whether the two measures of emotion regulation ability predict outcomes independently. Our hypotheses are supported for outcomes of relationship quality and academic achievement, but not for affective well-being outcomes. We discuss the implications for assessment of emotion regulation ability and the nature of outcomes predicted by emotion regulation ability.

## 1. Introduction

Emotion regulation ability predicts academic success, quality of interpersonal relationships, and well-being [1,2,3,4,5]. Someone who is able to regulate their anxiety during an important exam is likely to score better than a peer who is overwhelmed by anxiety. Similarly, an individual with greater ability to manage their irritation at a friend will be more successful at solving interpersonal problems and maintain more successful relationships.

Emotion regulation is the process of monitoring and adjusting emotional reactions to achieve a goal [6,7]. Emotion regulation can be conceptualized as a typical performance or a maximal performance attribute [1]. Defined as typical performance, it describes the frequency with which people use different (relatively more productive or unproductive) emotion regulation strategies in their daily lives and is measured through self-report instruments (e.g., expressive suppression strategy: “I control my emotions by not expressing them”, cognitive reappraisal strategy: “I control my emotions by changing the way I think about them” [8]). In this paper, we adopt the definition of emotion regulation as a maximal performance attribute—an individual’s ability to evaluate the effectiveness of different emotion regulation strategies. Maximal performance emotion regulation is measured through ability tests, which present people with descriptions of hypothetical emotion-laden situations and ask them to judge how useful different strategies and actions would be to achieve specific goals (i.e., how helpful would it be not to mention one’s sadness to a friend who is moving away if one has a goal of maintaining a friendship).

Recent research points toward observer reports as an important source of information about emotion regulation [9]. These reports involve asking a knowledgeable acquaintance, peer, family member, or friend to rate a target individual’s emotion regulation ability. While performance-tests are comprehensive measures of emotion regulation ability—the potential to adequately evaluate effectiveness of regulation strategies—they do not measure enacted emotion regulation in real-world situations. Observer reports of emotion regulation provide information about an individual’s enacted emotion regulation. Existing studies show that observer reports of emotion abilities predict higher effectiveness of graduate students as leaders and teammates, and supervisor ratings of emotional intelligence predict individual job performance and leadership effectiveness [9]. In the present study, we assess both performance-tested emotional regulation ability and observer reports of emotion regulation in a sample of high school students and test how these emotion regulation variables predict academic, interpersonal, and affective well-being outcomes.

### 1.1. ERA and Prediction of Life Outcomes

The present study examines emotion regulation ability (ERA) as a component of emotional intelligence—ability to successfully evaluate the effectiveness of emotion regulation strategies toward different goals [10]. This ability is assessed through performance-tests which ask respondents to judge how helpful different actions would be in a particular situation and toward achieving a particular goal (e.g., reaching an optimal level of anxiety or activation before a challenging task or reducing frustration in addressing a relationship conflict). For example, an ability test might describe a hypothetical situation where a protagonist is passed over for a promotion at work and faces the obstacle of maintaining a good relationship with their boss, as well as the coworker who received the promotion. Respondents evaluate the effectiveness of different actions for achieving this goal (e.g., how helpful it would be to focus on positive aspects of one’s current job).

Ability measures of emotion regulation are more closely related to measures of cognitive performance than personality, but are distinct from both [11]. This distinctiveness of ability measures of emotion regulation sets them apart from self-report measures, which ask respondents to evaluate their ability and behavior (e.g., ask them to judge how good they are at regulating emotions). People are poor judges of their emotion abilities and effectiveness of their behavior [11]. Self-reports of emotion abilities are not significantly related to performance on ability tests or show very low correlations with them. Furthermore, the predictive power of self-report measures of emotion regulation ability is reduced or becomes not significant when controlling for personality traits or psychological well-being. As a result of this, we do not employ self-reported measures to assess emotion regulation ability or associated behavior.

Performance measured ERA predicts achievement, relationship, and well-being outcomes. In adult samples, ERA predicts job satisfaction and job performance [12,13]. In child and adolescent samples, ERA predicts academic success. A recent meta-analysis of 162 studies found a significant relationship between ERA and academic performance measured through grade point average (GPA) and standardized test scores, even after controlling for cognitive ability and personality traits [5]. Other research shows that high school students’ ERA predicts school success as indicated by multiple measures beyond GPA—obtaining higher academic honors and recognitions, having fewer rule violations, as well as higher faculty-rated work ethic and citizenship, even after controlling for Grit and Conscientiousness, which are two personality traits associated with academic performance [1,14].

Higher ERA predicts less antagonism, fewer arguments, and more positive interactions within college student friendships [15], as well as greater satisfaction with relationships with parents and close friends [4]. College students’ ERA is associated with self- and peer-reports of interpersonal sensitivity and prosocial tendencies—handling interpersonal problems, appropriately expressing emotions in social situations, and understanding other people’s emotions [11,16]. The ability to understand and regulate emotions in college students predicts fewer negative behaviors with friends and family, including measures of physical and verbal fighting [17].

ERA also predicts different aspects of individual well-being, although the story here might be more complex than for achievement and interpersonal outcomes, and it might potentially depend on the types of well-being examined. ERA might differently predict affective well-being (satisfaction with life or happiness) and psychological well-being (which includes multiple dimensions such as personal growth and purpose in life [11,18]). Some studies of affective well-being show that higher ERA is positively associated with context-specific measures, such as job satisfaction [12] and school satisfaction [1]. Evidence for ERA predicting more general affective well-being is mixed. Burrus and colleagues [18] found that college students’ ERA is related to more positive affect and less negative affect assessed using the day reconstruction method. Similarly, some research shows that ERA is positively associated with self-reports of subjective happiness (feelings of overall happiness and happiness relative to others [19]) and satisfaction with life [20] and negatively associated with perceived stress [21]. However, other studies did not show ERA predicting general affect or satisfaction with life [11,22,23].

As a measure of maximal performance, ERA assesses the potential to act in ways that successfully promote specific achievement, relationship, and well-being goals. However, ERA measures do not necessarily describe behavior; it is possible that one can successfully evaluate regulation strategies when presented with them explicitly in hypothetical scenarios, but not commonly generate these strategies in daily situations on their own because of other influences on behavior (e.g., perceived utility of regulating emotions or personality traits that contribute to greater frequency of positive or negative affect). We propose that observer reports of emotion regulation can bridge the gap between potential (measured by ability tests) and enacted behavior.

### 1.2. Observer Reports of Emotion Regulation

Observer reports are measurements of traits, abilities, or behaviors as rated by colleagues, friends, family members, or acquaintances who are knowledgeable about a target individual (e.g., a teacher reporting on a student’s behavior). Substantial research documented agreement across different observers and self-observer agreement for judgments of social and emotional attributes [24,25,26,27]. Crucially, observer reports of social and emotional attributes predict important outcomes. For example, subordinate and peer reports of transformative leadership of undergraduate students (e.g., an individual places peer needs over their own needs) predict their appointment to leadership positions and performance evaluations of completed tasks assigned by supervisors [28]. In addition, observer-reports of affect by close friends and family members predict daily behaviors including laughing, crying, and social activity [29].

Research on observer reports of emotion regulation and other emotion abilities has been more scarce. Cross-judge agreement was supported for peer-reported abilities to express one’s own emotions and identify others’ emotions (e.g., being able to talk to others about experienced emotions or accurately describe the way others are feeling [30]). A multi-study analysis of reliability and validity of observer reports of emotional intelligence (which includes ERA) showed significant cross-rater agreement and distinctiveness of these reports from Big Five personality traits and emotion-related attributes such as liking and positive regard [9,31].

Existing research offers initial support for observer reports of emotional intelligence abilities predicting important outcomes. One meta-analysis found that controlling for cognitive ability and personality traits, observer-reported supervisor emotional intelligence predicts employee job satisfaction [12]. Furthermore, employee-reported supervisor emotional intelligence predicts reports of employee emotions, perceived opportunities for growth, and creativity at work provided by supervisors and employees [32,33]. Peer reports of graduate students’ emotional intelligence predict higher effectiveness as teammates and leaders in a collaborative group setting [9] and supervisor-reported transformative leadership and job performance [9]. In addition, peer-reported emotional intelligence is significantly related to classroom performance ratings of graduate students while controlling for liking and positive regard [9].

There are only a handful of studies of observer-reported emotion abilities in adolescent and young adult samples. One study found that undergraduate students’ GPA was predicted by observer-reports of emotional intelligence (provided by college roommates) beyond the Big Five traits and general mental ability [31]. In another study, parents of early adolescents completed a performance-based measure of ERA from the standpoint of their child (i.e., parents responded to questions in the way they imagined their children would respond [34]). These parent reports of children’s ERA predicted children’s school grades. However, the test that parents completed measured hypothetical ERA (projected potential) rather than emotion regulation observed through enacted behavior, which is the focus of the present study.

Some convergence was found between performance-tested emotion abilities and peer and supervisor ratings of these abilities [9,35]. However, the correlations between these measures were not high, suggesting that ability (assessed with performance-based tests) and observer ratings measure somewhat distinct constructs [9]. Since previous research has not addressed whether performance-tested emotion regulation ability and observer ratings of emotion regulation jointly and uniquely predict important outcomes, the goals of the present study are two-fold. Our first research question examines whether emotion regulation variables predict academic, interpersonal, and affective well-being outcomes in high school students beyond Big Five personality traits and gender. Hypothesis 1, based on reviewed research, is that emotion regulation variables have incremental validity over personality traits and gender. The second research question asks whether performance-tested ability and observer reports of emotion regulation independently predict outcomes. We propose that these two measures of emotion regulation provide distinct information and hypothesize that they each contribute unique explanation of examined outcomes.

## 2. Materials and Methods

### 2.1. Participants

Participants were 220 students (115 identified as male, 99 as female, 4 as transgender, 2 did not report gender) attending a private high school in the northeast of the United States (median age = 17). The school is organized into 6 teams, including one 9th grade team: (N = 37), two 10th grade teams (N = 32 and 40), two 11th grade teams (N = 26 and 28), and one 12th and college preparatory year team (N = 55). Most students in the sample came from middle-class family backgrounds (83.4% mothers and 82.1% of fathers earned college degrees or beyond). Students self-identified as 74.3% White/Caucasian, 13.3% Asian-American, 4.1% Black/African-American, 3.7% Hispanic, 2.8% mixed race, and 1.8% other.

### 2.2. Measures

#### 2.2.1. Emotion Regulation Ability

ERA was measured using the subtest from the Mayer, Salovey, and Caruso Emotional Intelligence Test—Youth Version [36,37]. The ERA subtest measures one’s capacity to evaluate how effective different actions would be toward achieving a specific regulation goal. The test presents six emotion-filled hypothetical scenarios describing everyday experiences relevant to adolescents in which they have to influence their emotional states (e.g., Li is excited about an upcoming party but has to study for an exam she has the next day). Each scenario is followed by 3 potential actions, and the respondent is asked to judge how effective each action would be toward achieving a specific goal (e.g., making Li study for the exam). The test is scored using expert scoring—responses are compared to judgments of emotion researchers guided by empirical research on effective emotion regulation strategies [37,38]. Test scores are standardized to have a mean of 100 and a standard deviation of 15.

#### 2.2.2. Observer Reports of Emotion Regulation

Observer reports of emotion regulation were obtained through teacher nominations. Teachers were presented a list of students they taught in alphabetical order and asked to select up to 5 students who they considered being on the low and high end of emotion regulation ability; they were asked to choose students from the list who were “good at regulating their emotions (e.g., able to get energized when necessary and calm down when needed)” and those “who have most difficulty regulating their emotions (e.g., not able to calm down when overwhelmed or angry, not able to get energized when necessary)”. Teachers were asked to think of what they have observed in their classes and not what they know about students in general (e.g., students’ reputation). Nominations of having difficulty regulating emotions were assigned the value −1 and nominations of being good at regulating emotions were assigned the value 1; students who were not nominated were assigned the value 0. To account for unequal number of students taught by different teachers, nominations by each teacher were standardized and summed across all teachers.

#### 2.2.3. Academic Outcomes

Academic outcomes included two measures: year-end grade point average (GPA) and academic honors. Students’ GPA was an average of GPAs over three academic terms and was provided by the school from official student records.

The school uses three levels of Latin honors: Cum Laude, Magna Cum Laude, and Summa Cum Laude. These honors are based on three sets of criteria: (1) grades (90 or higher average), (2) level of courses taken (e.g., number of advanced placement courses), and (3) academic citizenship behavior (teacher, coach, and community life leaders ratings of students’ modeling appropriate behavior, showing respect, and advocating responsibility). Scores were computed by assigning 1 point for each Cum Laude honor, 2 points for Magna Cum Laude, and 3 points for Summa Cum Laude and summing the points across 3 academic terms. Thus, the range of possible scores was between 0 (student has not achieved any academic honor) and 9 (student has achieved Summa Cum Laude honors each academic term).

#### 2.2.4. Interpersonal Relationship Outcomes

Interpersonal relationship outcomes were measured using peer nominations. Students were asked to nominate approximately 10% of their classmates who they would describe as (1) very understanding of how others feel and able to comfort them and (2) having the most leadership potential. The raw number of nominations was standardized within a team.

#### 2.2.5. Affective Well-Being Outcomes

Affective well-being was measured using two self-report scales, the Subjective Happiness Scale [39] and the School Satisfaction subscale of the Multidimensional Student Life Satisfaction Scale [40]. The Subjective Happiness Scale includes four items assessing overall happiness and asks respondents to compare their level of happiness to those of others (e.g., “Some people are generally very happy. They enjoy life regardless of what is going on, getting the most out of everything. To what extent does this characterization describe you?” [40], α = 0.80). Participants indicated their ratings on a 7-point scale.

In addition to overall happiness, we assessed satisfaction with a specific aspect of life. School Satisfaction scale included 8 items, which participants rated on a 6-point scale (e.g., “I look forward to going to school”, “I enjoy school a lot”, [40], α = 0.85).

#### 2.2.6. Big Five Personality Traits

The Big Five personality traits were measured using the self-report Big Five Inventory adapted for use with adolescent samples [41]: Extraversion (e.g., “is outgoing, sociable”; α = 0.83), Agreeableness (e.g., “is helpful and unselfish with others”; α = 0.74), Conscientiousness (e.g., “does things carefully and completely”; α = 0.78), Neuroticism (e.g., “worries a lot”; α = 0.77), and Openness to Experience (e.g., “is original, comes up with new ideas”; α = 0.78). Students rated each item on a 5-point scale.

### 2.3. Procedure

All measures were collected through a survey administered on the Qualtrics platform to groups of 10–15 students in classrooms. The measures were collected as a part of a larger study of social and emotional development of adolescents. Teacher nominations were collected using a Qualtrics survey during an all faculty meeting.

## 3. Results

Table 1 displays correlations between all study variables: Big Five personality traits, gender, emotion regulation variables, and academic, interpersonal, and well-being outcomes. ERA was positively correlated with Extraversion (r = 0.21, *p* = 0.005), Agreeableness (r = 0.43, *p* < 0.001), Conscientiousness (r = 0.29, *p* < 0.001), Openness (r = 0.30, *p* < 0.001), and gender (r = 0.25, *p* < 0.001; higher scores for female students). Teacher-observed emotion regulation was negatively correlated with Extraversion (r = −0.14, *p* = 0.040) and Neuroticism (r = −0.15, *p* = 0.029), and positively correlated with Agreeableness (r = 0.25, *p* < 0.001) and Conscientiousness (r = 0.22, *p* = 0.001). As expected, ERA and teacher-observed emotion regulation were significantly correlated, although the correlation was rather low (r = 0.22, *p* = 0.001).

ERA was significantly correlated with both academic outcome measures (GPA: r = 0.28, *p* < 0.001 and academic honors: r = 0.28, *p* < 0.001) and both interpersonal relationship outcomes (understanding others: r = 0.24, *p* < 0.001 and leadership potential: r = 0.25, *p* < 0.001). ERA had a positive correlation with school satisfaction (r = 0.29, *p* < 0.001) but was not significantly correlated with happiness (r = 0.12, *p* = 0.088).

Similar to performance-measured ERA, teacher-observed emotion regulation was correlated with both academic outcomes (GPA: r = 0.33, *p* < 0.001 and academic outcomes: r = 0.38, *p* < 0.001) and both interpersonal outcomes (understanding others: r = 0.19, *p* = 0.005 and leadership potential: r = 0.20, *p* = 0.003). Teacher-observed emotion regulation was positively correlated with school satisfaction (r = 0.19, *p* = 0.006) but not with happiness (r = −0.02, *p* = 0.748).

Next, we performed a series of hierarchical multiple regression analyses to examine the predictive validity of emotion regulation variables—performance-tested ERA and teacher-observed emotion regulation—for academic, interpersonal, and well-being outcomes (two variables assessing each). Hypothesis 1 posits that emotion regulation variables would predict academic, interpersonal, and well-being outcomes beyond the Big Five personality traits and gender. Hypothesis 2 states that performance-tested ERA and teacher-observed emotion regulation would independently predict the outcomes.

Before running these analyses, we examined the assumptions of linear regression. Examination of scatter plots showed that study variables were linearly related. The two variables of social outcomes (peer nominations of leadership potential and understanding others) were not normally distributed (but positively skewed). Violations of normality reduce the size of regression coefficients [42]. The current study does not focus on the magnitude of effects but rather on whether two measures of emotion regulation ability are independent predictors of outcomes. Furthermore, simulations studies show that violations of normality do not have a substantial effect on validity of regression results [43]. Thus, showing support for emotion regulation variables predicting outcomes would adequately address the research questions.

Homoscedasticity assumption was supported for academic and well-being outcomes but was violated for the social outcomes. Monte Carlo simulation studies [43] show that the risk of type I error increases with violations of homoscedasticity. However, although this bias can be detected in simulations, the type I error levels increase from 0.05 to approximately 0.06, suggesting caution in interpreting *p*-values close to 0.05.

The Big Five personality traits and gender were entered in Step 1 of each regression. Step 2 added emotion regulation variables (see Table 2). Two regressions tested emotion regulation variables predicting academic outcomes: year-end GPA and academic honors. Step 1 variables significantly predicted GPA (Δ$R^{2}$ = 0.168, *p* < 0.001) and academic honors (Δ$R^{2}$ = 0.171, *p* < 0.001). Extraversion was a significant Step 1 predictor for both variables (GPA: β = −0.16, *p* = 0.024; academic honors: β = −0.19, *p* = 0.006), as well as Conscientiousness (GPA: β = 0.33, *p* < 0.001; academic honors: β = 0.29, *p* < 0.001) and gender (GPA: β = 0.21, *p* = 0.004; academic honors: β = 0.21, *p* = 0.002). Openness was also a significant predictor of GPA in Step 1 (β = 0.14, *p* = 0.04). In Step 2, in addition to Conscientiousness and gender, emotion regulation variables predicted both academic success variables, supporting Hypothesis 1 (Δ$R^{2}$ = 0.084, *p* < 0.001, and Δ$R^{2}$ = 0.105, *p* < 0.001, for GPA and academic honors, respectively). Furthermore, academic outcomes were independently predicted by both performance-tested ERA (GPA: β = 0.18, *p* = 0.02; academic honors: β = 0.17, *p* = 0.018) and teacher-reports of student emotion regulation (GPA: β = 0.26, *p* < 0.001; academic honors: β = 0.30, *p* < 0.001). These findings are consistent with Hypothesis 2.

Two regressions predicted interpersonal outcomes: peer nominations of ability to understand others and leadership potential. Step 1 variables significantly predicted peer nominations of understanding others (Δ$R^{2}$ = 0.106, *p* = 0.001; Extraversion: β = 0.16, *p* = 0.027, and gender: β = 0.26, *p* < 0.001) but not peer nominations of leadership potential. As predicted by our first hypothesis, emotion regulation variables significantly predicted understanding others, Δ$R^{2}$ = 0.063, *p* = 0.001, and leadership potential, Δ$R^{2}$ = 0.076, *p* < 0.001. Performance-tested ERA and teacher-observed emotion regulation were significant independent predictors for understanding others (β = 0.16, *p* = 0.029 and β = 0.21, *p* = 0.003, respectively) and leadership potential (β = 0.22, *p* = 0.005, and β = 0.20, *p* = 0.006), supporting again Hypothesis 2.

The last two regressions predicted affective well-being outcomes— subjective happiness and satisfaction with school. In Step 1, subjective happiness was predicted by Extraversion, β = 0.36, *p* < 0.001, Agreeableness, β = 0.21, *p* = 0.002, and Neuroticism, β = −0.28, *p* < 0.001, and satisfaction with school was predicted by Conscientiousness, β = 0.31, *p* < 0.001, Neuroticism, β = −0.25, *p* < 0.001, and gender β = 0.13, *p* = 0.049. Step 2 variables did not significantly predict affective well-being outcomes (subjective happiness: Δ$R^{2}$ = 0.003, *p* = 0.616; school satisfaction: Δ$R^{2}$ = 0.022, *p* = 0.065); thus, Hypothesis 1 was not supported). School satisfaction was significantly predicted by performance ERA, β = 0.16, *p* = 0.027 but not teacher observations of emotion regulation, β = 0.04, *p* = 0.603. Thus, Hypothesis 2 was not supported.

## 4. Discussion

We tested the predictive validity of two emotion regulation variables—performance-tested individual ERA and observer-reported emotion regulation—on academic, interpersonal, and affective well-being outcomes. The hypotheses stated that emotion regulation variables would predict outcomes beyond the effects of Big Five personality traits and gender and that performance-tested ERA and observer reports of emotion regulation would be independent predictors of these outcomes. Results showed that emotion regulation variables significantly and independently predicted academic and interpersonal outcomes, but not affective well-being outcomes.

To evaluate the role of emotion regulation across different classes of outcomes, two specific measures were selected to address each. The conclusions that can be drawn depend on the nature of these outcomes. Academic outcomes were assessed using year-end GPA and achievement of academic honors. GPA has been the most commonly used measure of academic success in past studies [5] and was included in the study because of its practical importance (e.g., in contributing to college entrance) and for the purpose of comparison with previously published results. Academic honors are less commonly used in research, but they are conceptually important because they indicate a high level of achievement and aspiration (as a measure taking into account grades, but also assessments of academic motivation and academic citizenship/responsibility). The present research provides evidence for the independent contribution of performance-tested ERA and observed emotion regulation to the prediction of school success.

Interpersonal relationship outcomes were assessed using peer nominations of understanding how others feel and peer nominations of leadership potential. Previous research attests to the validity of these measures. Self-observer agreement has been established for judgments of interpersonal relationships [44,45], and significant inter-rater agreement has been shown for peer reports of adolescent interpersonal sensitivity [25]. Results of the present study echo previous research. ERA was found to predict the quality of social interaction measured by peer reports of interpersonal sensitivity (e.g., reports on items such as, “Is this person sensitive to the feelings or concerns of other people?” [15]). Performance-tested ERA also predicts peer nominations of leadership potential (e.g., “This person provided inspiring strategic and group goals.” [9]). Moreover, leadership potential has been examined in relation to peer-reported emotion regulation, enabling us a comparison with limited previous research of observer-reported emotion abilities [9]. The present study extends previous research on performance-tested ERA and adds the finding that observed emotion regulation can independently predict significant interpersonal outcomes.

The choice of well-being measures was especially challenging, considering the breadth of the conceptual domain of well-being that includes hedonic or affective well-being and eudaimonic or psychological well-being [46,47,48]. Rather than examine a single criterion pertaining to each of these two major types of well-being, we opted to assess only affective well-being using two measures—overall subjective happiness and a domain specific satisfaction with school. Previous research offered some support for the role of performance-tested ERA in relation to affective well-being (happiness [19]; perceived stress [21]; satisfaction with school [1]). However, evidence for the role of performance-tested ERA in affective well-being is not unequivocal (e.g., significant relationship not found by [11,22,23]).

The results of the present study offered limited support for the role of emotion regulation in affective well-being. Satisfaction with school was predicted by performance-tested ERA, but teacher-observed emotion regulation did not add to the prediction. It is possible that the beneficial effects of emotion regulation for academic success spill to the affective variable reflecting such academic success—satisfaction with school. However, results did not provide support for the role of emotion regulation in overall subjective happiness. This measure is trait-like in nature; it describes a general evaluation of oneself as a happy person and a judgment of happiness relative to one’s peers. As such, this measure is similar to trait positive affectivity. Thus, it is not surprising that it was predicted by trait Extraversion (which includes positive emotionality) and Neuroticism (which includes negative emotionality [49]). Trait-like measures of affective well-being might be obscuring the role of emotion regulation in influencing affect at specific times and situations.

The novel aspect of the present study is the examination of observer reports of emotion regulation. Observer reports of emotion regulation have not been extensively studied, especially in comparison to performance-tested ERA. Song and colleagues [31] showed that observer-reported emotional intelligence predicts academic performance, although they did not compare independent contributions of performance-tested and observer-reported emotion abilities. In university student samples, observer-reported emotion regulation predicted academic performance, quality of social interaction, and peer ratings of performance [9]. The present study contributed to our understanding of emotion regulation by showing that both performance-tested and observer-reported emotion regulation independently contribute to relevant outcomes. This is important practically and theoretically. Practically, these results speak to the usefulness of employing observer reports of emotion abilities for predicting outcomes of interest, which is especially pertinent when performance-tested assessments are not possible. Theoretically, these results pose a question about what kind of information observers use to make their judgments about emotion regulation ability, as well as questions about what factors in addition to ability contribute to behavior that can be perceived as successful emotion regulation.

Behavior indicating successful emotion regulation is in part based on one’s potential to imagine and reason about actions that could be effective in emotion-laden situations (which is assessed by performance measures). However, emotion regulation that can be observed in behavior is also in part influenced by other non-ability variables, such as motivation for regulating emotions toward different goals or in different contexts. Expectancy-value theories [50] describe behavior as based on motivational variables of beliefs—self-evaluation whether one can be successful in a certain task, and the extent to which they perceive value in it. Applied to regulating emotions, enacting regulation will depend on motivational variables of perceived efficacy (e.g., Can I regulate my feelings at school?) and values of regulation (e.g., Will it be useful for me to regulate emotions at school? Is it important to regulate emotions at school?). Self-reported evaluations of one’s own emotion regulation ability are similar to affective self-regulatory efficacy [51], opening an important avenue for research on contributors to successful regulatory behavior as influenced by ability and motivational variables.

While this was the first study to examine the predictive power of performance-tested ERA and observer reports of emotion regulation for a host of outcomes, there were several limitations to our research. The sample size was relatively small and consisted of students attending the same private high school who came from middle-class family backgrounds. Due to the restrictions of student socio-economic status and diversity on our sample, our findings may not be fully representative of the greater adolescent population.

Control variables in the present study were selected to be most relevant for the whole set of examined outcomes. This is why we focused on Big Five personality traits and gender (Conscientiousness predicting academic outcomes: [52]; Extraversion, Conscientiousness, and Neuroticism predicting well-being: [53]; Extraversion and Agreeableness predicting quality of social relationships: [2]; emotion regulation ability being higher in women: [11]). Another important control variable would have been general cognitive ability, which is related to the performance-tested emotion regulation ability and at least one of the examined outcomes, academic achievement [36]. A recent meta-analysis shows that emotion regulation ability predicts academic achievement after controlling for both cognitive ability and personality traits [5]. Although the lack of a measure of cognitive ability is a limitation of this study, existing research (including the meta-analysis) supports the idea that the results are not simply due to the overlap between emotion regulation ability and cognitive ability.

Another limitation of this study was our measurement of observer-reports. The measure of teacher-observed emotion regulation based on a single question; teachers selected students who were either good or poor at regulating emotions in the classroom. Ideally, observer-reported emotion regulation would include multiple questions about instances of students regulating emotions at school. Our findings support observer-reported emotion regulation as a predictor of life outcomes, suggesting that future research should be conducted with multi-item (and therefore more reliable) measures of emotion regulation.

Understanding others’ feelings and leadership potential—two interpersonal relationship outcomes examined in this study—were both measured through peer nominations. This might be seen as a limitation because if valid, these ratings yield by necessity positively skewed data. For instance, if students agree about which peers have the most leadership potential, most students will receive few nominations, and a minority will receive multiple nominations. However, peer nominations are relevant measures in this case because the quality of interpersonal relationships is influenced by how others see a person; thus, peer opinions can constitute the most important interpersonal outcomes. A more comprehensive approach to assessing interpersonal outcomes would include both self-reports and peer nominations of quality of social interactions [4]. Such comprehensive assessments would enable researchers to draw conclusions about which outcomes are most closely related to emotion regulation. For instance, one study found that ERA predicted peer nominations of interpersonal sensitivity but not self-reported interpersonal sensitivity [4]. Would observer-reported emotion regulation predict both self- and peer-reported outcomes?

Including subjective happiness as an outcome of well-being was a possible limitation as well; we chose to study hedonic well-being rather than psychological well-being. While the results of the present study offer limited support for the role of emotion regulation in affective well-being, we cannot make conclusions about well-being more broadly. Indeed, two studies that examined both psychological and affective well-being suggest that emotion abilities are more relevant to former than the latter. Brackett and Mayer found that overall emotional intelligence predicted psychological well-being but not subjective well-being [11]. Specifically measuring ERA, Burrus and colleagues [18] found a stronger relationship with psychological well-being than measures of daily affect. Future studies should examine psychological well-being and its relationship with emotion regulation. Psychological well-being includes facets such as personal growth, purpose in life, and a sense of mastery in dealing with life challenges [47,48], which are all outcomes that could be predicted by an individual’s ability to successfully regulate emotions toward goals of personal learning and development.

## 5. Conclusions

The present results point to the need to understand emotion regulation in terms of both one’s ability—the potential to regulate emotions successfully—and observed behavior—indicators of successful emotion regulation—when trying to predict academic and interpersonal outcomes in adolescents. Since performance-tested ERA and observer-reported emotion regulation are measuring distinct attributes of potential and enacted behavior, we need to better understand which outcomes are jointly predicted and which are not. This research directs us to further examine what might go into the enacting of behavior beyond the ability to reason about emotion regulation strategies. Future research should explore these potential variables further, including the mindset about the importance of attending to emotions and the perceived utility of emotion regulation for achieving significant outcomes.

## Figures and Tables

**Table 1 ijerph-18-03204-t001:** Zero-order correlations among all study variables.

	Big Five Personality Traits					Gender	Emotion Regulation		Academic Outcomes		Interpersonal Outcomes		Well-Being Outcomes	
	1.	2.	3.	4.	5.	6.	7.	8.	9.	10.	11.	12.	13.	14.
Big Five														
1. Extraversion	—													
2. Agreeableness	0.27 ***	—												
3. Conscientiousness	0.07	0.35 ***	—											
4. Neuroticism	−0.24 ***	−0.27 ***	−0.12	—										
5. Openness	0.25 ***	0.20 **	0.15 *	−0.08	—									
6. Gender	0.15 *	0.19 **	0.08	0.14 *	0.08	—								
Emotion regulation														
7. ERA	0.21 **	0.43 ***	0.29 ***	−0.09	0.30 ***	0.25 ***	—							
8. Teacher report ER	−0.14 *	0.25 ***	0.22 **	−0.15 *	0.13	0.11	0.22 **	—						
Academic outcomes														
9. Total GPA	−0.08	0.09	0.32 ***	−0.01	0.14 *	0.18 *	0.28 ***	0.33 ***	—					
10. Academic honors	−0.13	0.10	0.29 ***	0.09	0.10	0.22 **	0.28 ***	0.38 ***	0.75 ***	—				
Interpersonal outcomes														
11. PN: understanding others	0.20 **	0.06	0.08	−0.06	0.04	0.26 ***	0.24 ***	0.19 **	0.22 **	0.30 ***	—			
12. PN: leadership potential	0.10	0.05	0.17 *	0.01	0.06	0.10	0.25 ***	0.20 **	0.36 ***	0.44 ***	0.64 ***	—		
Well-being outcomes														
13. Happiness	0.46 ***	0.36 ***	0.08	−0.43 ***	0.08	−0.02	0.12	−0.02	−0.10	−0.11	0.11	0.04	—	
14. School satisfaction	0.09	0.24 **	0.37 ***	−0.27 ***	0.11	0.13	0.29 ***	0.19 **	0.28 ***	0.28 ***	0.09	0.13	0.15 *	—

Note: * *p* < 0.05, ** *p* < 0.01, *** *p* < 0.001. ERA—emotion regulation ability; ER—emotion regulation; PN—peer nominations.

**Table 2 ijerph-18-03204-t002:** Multiple regression variables predict academic, interpersonal, and well-being outcomes while controlling for Big Five.

	*Academic Outcomes*						*Interpersonal Outcomes*						*Well−Being Outcomes*					
	Total GPA			Academic Honors			PN: Understanding Others			PN: Leadership Potential			Happiness			School Satisfaction		
	β	95% CI		β	95% CI		β	95% CI		β	95% CI		β	95% CI		β	95% CI	
		Lo	Up		Lo	Up		Lo	Up		Lo	Up		Lo	Up		Lo	Up
Step 1	$\Delta R^{2}$ = 0.168 ***			$\Delta R^{2}$ = 0.171 ***			$\Delta R^{2}$ = 0.106 **			$\Delta R^{2}$ = 0.045			$\Delta R^{2}$ = 0.351 ***			$\Delta R^{2}$ = 0.197 **		
Extraversion	−0.16 *	−0.15	−0.01	−0.19 **	−10.26	−0.22	0.16 *	0.03	0.45	0.09	−0.09	0.37	0.36 ***	0.35	0.72	−0.03	−0.23	0.15
Agreeableness	−0.11	−0.16	0.03	−0.02	−0.81	0.57	−0.09	−0.44	0.12	−0.07	−0.44	0.16	0.21 **	0.15	0.65	0.01	−0.23	0.27
Conscientiousness	0.33 ***	0.10	0.27	0.29 ***	0.69	10.87	0.08	−0.10	0.39	0.18 *	0.06	0.58	−0.04	−0.29	0.14	0.31 ***	0.28	0.71
Neuroticism	−0.05	−0.11	0.05	0.06	−0.32	0.81	−0.08	−0.37	0.10	0.03	−0.19	0.31	−0.28 ***	−0.65	−0.26	−0.25 ***	−0.57	−0.16
Openness	0.14 *	0.00	0.17	0.09	−0.18	10.01	−0.04	−0.32	0.17	0.02	−0.23	0.29	−0.07	−0.33	0.10	0.05	−0.14	0.28
Gender	0.21 **	0.05	0.25	0.21 **	0.44	10.87	0.26 ***	0.27	0.86	0.07	−0.15	0.48	−0.07	−0.40	0.11	0.13 *	0.00	0.52
Step 2	$\Delta R^{2}$ = 0.084 ***			$\Delta R^{2}$ = 0.105 ***			$\Delta R^{2}$ = 0.063 **			$\Delta R^{2}$ = 0.076 ***			$\Delta R^{2}$ = 0.003			$\Delta R^{2}$ = 0.022		
Extraversion	−0.09	−0.12	0.03	−0.11	−0.95	0.07	0.21 **	0.10	0.53	0.14	−0.02	0.44	0.34 ***	0.32	0.71	−0.03	−0.23	0.16
Agreeableness	−0.22 **	−0.23	−0.04	−0.13	−10.29	0.07	−0.17 *	−0.61	−0.04	−0.17*	−0.64	−0.03	0.23 **	0.17	0.70	−0.04	−0.34	0.18
Conscientiousness	0.27 ***	0.07	0.23	0.23 **	0.44	10.57	0.03	−0.18	0.30	0.12	−0.03	0.48	−0.03	−0.28	0.16	0.29 ***	0.24	0.67
Neuroticism	−0.03	−0.09	0.06	0.10	−0.14	0.94	−0.05	−0.32	0.14	0.06	−0.14	0.35	−0.29 **	−0.67	−0.27	−0.24 **	−0.56	−0.16
Openness	0.07	−0.04	0.13	0.02	−0.47	0.68	−0.10	−0.42	0.07	−0.05	−0.35	0.17	−0.05	−0.32	0.12	0.01	−0.20	0.23
Gender	0.16 *	0.02	0.21	0.15 *	0.15	10.53	0.21 **	0.16	0.74	0.02	−0.28	0.35	−0.06	−0.39	0.15	0.10	−0.06	0.46
ERA	0.18 *	0.00	0.01	0.17 *	0.01	0.05	0.16 *	0.00	0.02	0.22 **	0.01	0.03	−0.02	−0.01	0.01	0.16 *	0.00	0.02
Teacher report ER	0.26 ***	0.07	0.22	0.30 ***	0.70	10.78	0.21 **	0.12	0.57	0.20 **	0.10	0.58	−0.06	−0.30	0.11	0.04	−0.15	0.26
Final model	$R^{2}$ = 0.252			$R^{2}$ = 0.276			$R^{2}$ = 0.169			$R^{2}$ = 0.121			$R^{2}$ = 0.354			$R^{2}$ = 0.219		
	F(8,193) = 70.78 ***			F(8, 211) = 90.66 ***			F(8, 212) = 50.19 ***			F(8, 212) = 30.51 **			F(8, 201) = 130.23 ***			F(8, 204) = 60.87 ***		

Note: * *p* < 0.05, ** *p* < 0.01, *** *p* < 0.001. ERA—emotion regulation ability; ER—emotion regulation; PN—peer nominations.

## Data Availability

The data presented in this study are available on request from the corresponding author. The data are not publicly available due to third party restrictions.

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
