# Peer review of "Emotion Regulation Ability: Test Performance and Observer Reports in Predicting Relationship, Achievement and Well-Being Outcomes in Adolescents"

_ijerph, 2021, doi:10.3390/ijerph18063204_

Round 1

Reviewer 1 Report

Once again, I am ready to comment on the considerations that I deem appropriate to make in the manuscript so that its publication can be considered. I do not know if due to the short period of time, some of the recommendations I address to the authors are not met with the depth they deserve.

For example, there are still errors in references. This requirement seems basic to me. Please check carefully that the references contain all the necessary data and that they have been made correctly.

I am still concerned about the absence of a sufficient theoretical body that supports the need to carry out this study. Why is it important to study the motion regulation ability in adolescents? What consequences can it have? Why have not developed instruments for adverse events reported in adolescents? Although this request may be reiterative, I believe that this section needs to be significantly improved. 

The results are very eassy for the description of the sample and are not informative. 

Discussion and Conclusions sections are not informative. 

I dont believe this study adds a great deal of novel and new information.

Finally, I hope the outcome of this specific submission will not discourage you from the submission of future manuscripts.

Best wishes in all your future endeavors.

Author Response

Dear Reviewer 1,

Below we address each of your comments and suggestions along with the changes we made in the revised manuscript. 

Once again, I am ready to comment on the considerations that I deem appropriate to make in the manuscript so that its publication can be considered. I do not know if due to the short period of time, some of the recommendations I address to the authors are not met with the depth they deserve.

For example, there are still errors in references. This requirement seems basic to me. Please check carefully that the references contain all the necessary data and that they have been made correctly.

We have carefully checked the references and believe they are complete and correct.

I am still concerned about the absence of a sufficient theoretical body that supports the need to carry out this study. Why is it important to study the motion regulation ability in adolescents? What consequences can it have? Why have not developed instruments for adverse events reported in adolescents? Although this request may be reiterative, I believe that this section needs to be significantly improved. 

We start the introduction section by pointing that the emotion regulation ability needs to be studied because of its relationship with important outcomes, including well-being, achievement, and relationship quality. While this importance of emotion regulation ability is well documented in previous research, our study is one of the first to examine whether performance on an emotion regulation ability test and observations by knowledgeable others can independently predict life outcomes.

The discussion section acknowledges the limitations of the observer measure of emotion regulation and we call for future research to develop a comprehensive observational measure. 

The results are very eassy for the description of the sample and are not informative. 

Discussion and Conclusions sections are not informative. 

We addressed all the concerns raised in a previous round of review. Without specifics on which points were not addressed to the reviewer’s satisfaction, we are not able to make additional changes.

Reviewer 2 Report

I believe the manuscript has been improved and now warrants publication in the IJERPH.

Author Response

Dear Reviewer 2,

Thank you for your assessment. We are grateful that you find the revised manuscript suitable for publication.

Round 2

Reviewer 1 Report

Thank you very much for the new opportunity to evaluate the manuscript. And I thank the authors for their work and effort in answering my questions. However, I do maintain my positions, not for the sake of the study itself. But yes, due to the methodological limitation that makes the results found unfeasible.